# Neighbor Does Matter: Global Positive-Negative Sampling for Vision-Language Pre-training

## ABSTRACT

Sampling strategies have been widely adopted in Vision-Language Pre-training (VLP) and have achieved great success recently. However, the sampling strategies adopted by current VLP works are limited in two ways: i) they only focus on negative sampling, ignoring the importance of more informative positive samples; ii) their sampling strategies are conducted in the local in-batch level, which may lead to sub-optimal results. To tackle these problems, in this paper, we propose a Global Positive-Negative Sampling (GPN-S) framework for vision-language pre-training, which conducts both positive and negative sampling in the global level, grounded on the notion of neighborhood relationships. Specifically, our proposed GPN-S framework is capable of utilizing positive sampling to bring semantically equivalent samples closer, as well as employing negative sampling to push challenging negative samples farther away. We jointly consider them for vision-language pre-training on the global-level perspective rather than a local-level mini-batch, which provides more informative and diverse samples. We evaluate the effectiveness of the proposed GPN-S framework by conducting experiments on several common downstream tasks, and the results demonstrate significant performance improvement over the existing models.

## CCS CONCEPTS

• **Computing methodologies → Artificial intelligence**.

## KEYWORDS

Vision-Language Pre-training, Positive-Negative Sampling

## 1 INTRODUCTION

In recent years, there has been a burgeoning interest in the field of vision-language pre-training (VLP) within the AI community[2, 10, 19, 25, 32]. VLP aims to learn multimodal representations from large-scale image-text pairs that can simultaneously process visual and textual data, with the goal of improving performance on downstream tasks such as image-text retrieval and visual question answering.

Multi-modal alignment is fundamental to the efficacy of VLP, where semantically similar samples are mapped together to enhance representation accuracy. Recent studies[19, 25] have demonstrated that intelligent sampling strategies can significantly improve VLP

Permission to make digital or hard copies of all or part of this work for personal or classroom use is granted without fee provided that copies are not made or distributed for profit or commercial advantage and that copies bear this notice and the full citation on the first page. Copyrights for components of this work owned by others than the author(s) must be honored. Abstracting with credit is permitted. To copy otherwise, or republish, to post on servers or to redistribute to lists, requires prior specific permission and/or a fee. Request permissions from permissions@acm.org.

*ACM MM, 2024, Melbourne, Australia*

© 2024 Copyright held by the owner/author(s). Publication rights licensed to ACM.
ACM ISBN 978-x-xxxx-xxxx-x/YY/MM
https://doi.org/10.1145/nnnnnnn.nnnnnnn

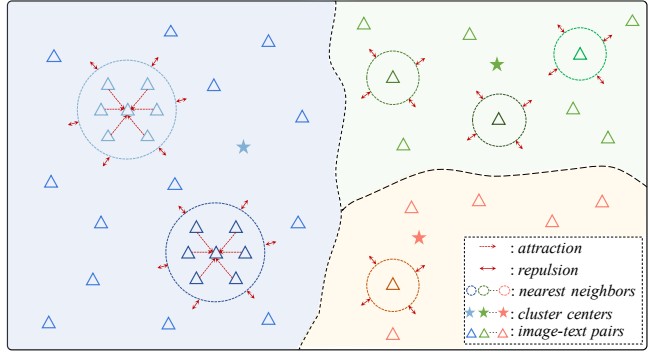

**Figure 1: Our proposed sampling strategy involves leveraging information from nearest neighbors to identify and sample semantically equivalent positives, as well as employing a cluster algorithm to obtain global-level challenging negatives.**

model performance. For instance, the CLIP[25] model has shown the benefits of using large mini-batch sizes to obtain hard negatives, thereby increasing model effectiveness. Similarly, ALBEF[19] uses a large negative queue to extract challenging negative examples, leading to notable performance improvements. These developments highlight the potential of leveraging informative samples for further advancements. However, two primary limitations persist:

- Current works predominantly focus on sampling hard negatives within mini-batches[2, 19, 32], a local-level approach that yields less optimal negative samples.
- There is limited research on positive sampling strategies, with many studies overlooking the potential of positive sampling in enriching multi-modal alignment information.

To overcome these limitations, we propose the Global Positive-Negative Sampling (GPN-S) framework for VLP. This innovative framework conducts both positive and negative sampling on a global level, moving beyond the constraints of local-level mini-batches. Specifically, GPN-S maps the entire dataset's samples into a unified embedding space and utilizes their neighborhood relationships to determine which samples should be closer or farther apart, as illustrated in Figure 1. This approach provides more informative and diverse samples. For global positive sampling (GP-S), we aggregate co-occurring nouns from neighboring texts, replace noisy web-crawled text to enhance the model's cross-modal alignment ability. We also mine semantically equivalent images to improve the model's uni-modal alignment ability. For global negative sampling (GN-S), we employ a global cluster technique to obtain more challenging negative samples. Notably, our method is model-agnostic and can be applied to enhance most existing VLP models.

In brief, our contributions can be summarized as follows:

- We propose the GPN-S framework, a novel approach that broadens the sampling strategy to encompass both positive and negative pairs at a global level, thus significantly enhancing VLP model performance.
- We utilize the relationships between neighboring samples to effectively integrate information from the embedding space, improving the quality of representations in both cross-modal and uni-modal scenarios.
- We conduct extensive experiments on several downstream tasks to demonstrate that our GPN-S can significantly yield better performance than the models trained without GPN-S.

## 2 RELATED WORKS

### 2.1 Vision and Language Pre-training

Vision-Language Pre-training (VLP) models are primarily categorized into two types: dual encoder models and fusion encoder models. Dual encoder models, such as CLIP[25] and ALIGN[13], independently encode images and texts, employing contrastive learning to align image-text pair embeddings. While effective in retrieval tasks, their limited interaction between modalities restricts performance in complex tasks like Visual Question Answering (VQA).

Fusion encoder models overcome this by integrating a fusion encoder to meld features from uni-modal encoders. Early iterations[8, 20, 22, 29] relied on pre-trained object detectors for visual feature extraction, facing significant time overhead and capacity constraints. ViLT[16] addressed this by directly inputting image patch features and text token embeddings into a ViT[9] model, but this approach lagged behind object-detector based models due to inadequate uni-modal modeling. Subsequent developments, such as ALBEF[19], combined the benefits of dual and fusion encoders, inspiring further enhancements in models like TCL[32]. TCL introduced intra-modal and global-local contrastive losses to achieve better alignment capabilities.

Recent developments in multi-modal large language models have also been based on vision and language pre-training. For instance, BLIP-2[18] introduced a lightweight Q-Former to bridge the modality gap between vision and text. It employs a two-stage training process, requiring an initial stage of vision and language pre-training to learn how to align with text before connecting to the LLM in the second stage.

Our framework distinguishes itself by being model-agnostic, compatible with a broad range of VLP models. It uniquely enhances performance by refining the sampling strategies of these models, a critical factor not explicitly addressed in previous approaches.

### 2.2 Sampling for Positive and Negative

The primary focus of representation learning lies in extracting semantic information from data, where the representations of similar (positive) pairs are clustered together and those of dissimilar (negative) pairs are spread apart[3]. The sampling of positive and negative pairs is critical in the success of VLP alignment.

**Positive sampling** traditionally focuses on generating semantically similar pairs, particularly through random data augmentation in uni-modal contexts[6, 7, 30]. Approaches like NNCLR[11] have leveraged the nearest-neighbor sample as the positive. It leverages a queue for nearest-neighbor identification in uni-modal data, which

is notably trained on the 1000-class ImageNet dataset. In such a clean, categorized environment, a queue suffices for identifying similar neighbors. However, challenges emerge in web-crawled, noisier datasets where identifying semantically similar positive samples requires more nuanced strategies. This underscores the need for advanced positive sampling approaches in cross-modal scenarios, a domain less explored in previous research.

**Negative sampling** is aimed at identifying pairs with dissimilar semantics yet closely embedded representations. Techniques that incorporate hard negative sampling have shown to be beneficial. For instance, ALBEF[19] selects the nearest sample in a mini-batch as the hard negative. VLMo[2] extends this approach by mining hard negatives from a broader pool, gathering training examples across all GPUs, In contrast, GRIT-VLP[5] maintains a queue to identify hard negatives, resulting in more significant improvements.

Our work introduces a novel unified positive-negative sampling framework, distinct for its global-level approach. Unlike existing methods constrained by the limited scope of mini-batches or queues, our framework performs sampling across the entire dataset. This comprehensive strategy enables more effective differentiation and identification of positive and negative samples, especially in noisy, diverse datasets, addressing gaps left by previous VLP research and offering a significant advancement in the field.

## 3 METHOD

In this section, we present the Global Positive-Negative Sampling (GPN-S) framework, which is designed to improve the performance of existing VLP models by refining their sampling strategies. We will first introduce the preliminaries about how current VLP models align image and text.

### 3.1 Preliminaries

For a VLP model composed of a visual encoder $f_V(\cdot)$, a text encoder $f_T(\cdot)$, and a fusion encoder $f_F(\cdot, \cdot)$, the unimodal representations for an image-text pair $(v, t)$ are obtained as follows:

$$\{\mathbf{v}^{\text{cls}}, \mathbf{v}^1, \mathbf{v}^2, ...\} = f_V(v), \tag{1}$$

$$\{\mathbf{t}^{\text{cls}}, \mathbf{t}^1, \mathbf{t}^2, ...\} = f_T(t), \tag{2}$$

where $\mathbf{v}^{\text{cls}}$ and $\mathbf{t}^{\text{cls}}$ represent the embeddings of the corresponding [CLS] token. The fusion representation is obtained through $f_F(f_V(v), f_T(t))$.

Training the VLP model typically involves adopting the image-text matching objective, which introduces a predict head $f_H(\cdot)$ to predict the matching likelihood between the image and text. The probability of a match between $v$ and $t$ is calculated as:

$$p(v, t) = f_H(f_F(f_V(v), f_T(t))). \tag{3}$$

Given a batch of $N$ image-text pairs $\{(v_i, t_i)\}_{i=1}^{N}$, where $(v_i, t_i)$ is referred to as a positive pair and $(v_i, t_j), \forall j \neq i$ as a negative pair. The image-text matching loss on the positive pair $(v_i, t_i)$ is calculated using binary cross-entropy loss:

$$\mathcal{L}_{\text{itm}}(v_i, t_i) = -log(p(v_i, t_i)) - log(1 - p(v_i, t_j)), \tag{4}$$

where $t_j (j \neq i)$ is a randomly selected text from the batch. Recent works[2, 19] introduce hard negative sampling by selecting a $t_j$ to maximize the dot product $(\mathbf{v}_i^{\text{cls}})^T \mathbf{t}_j^{\text{cls}}$ within the batch. The hard

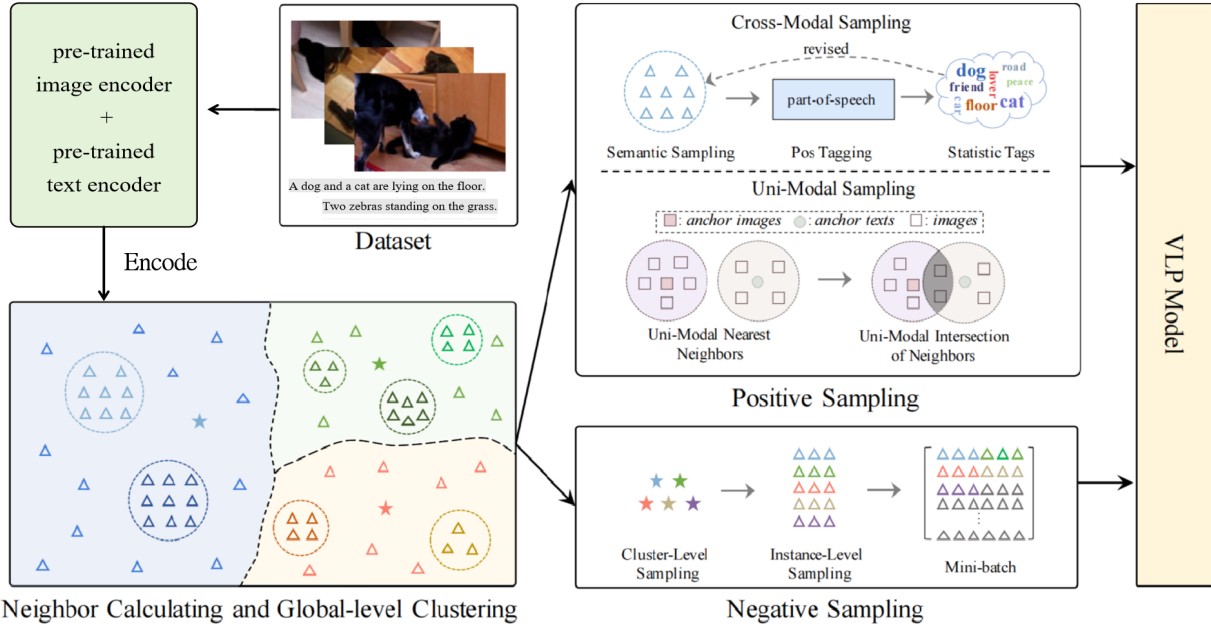

Figure 2: The proposed method contains two primary strategies: 1) Global Positive Sampling(GP-S), which aims to sample both cross-modal and uni-modal data pairs that share same semantic through the neighboring information. 2) Global Negative Sampling(GN-S), which aims to sample more challenging negative data through the clustering information.

negative pairs share similar semantics but differ in fine-grained details, thereby enhancing the model's ability for fine-grained understanding. However, the quality of hard negatives of existing works is constrained by the batch size $N$.

Different VLP models may introduce different loss functions, all of which are designed to align positive image-text pair $(v_i, t_i)$. Without loss of generality, we denote all other losses as $\mathcal{L}_{\text{other}}$, and the final training loss is expressed as:

$$\mathcal{L}_f(v_i, t_i) = \mathcal{L}_{\text{itm}}(v_i, t_i) + \mathcal{L}_{\text{other}}(v_i, t_i). \quad (5)$$

While current VLP models exhibit outstanding performance across diverse tasks, there are still some limitations in their sampling strategies for positive and negative pairs. Firstly, due to the prohibitive cost of manual labeling, researchers have to train VLP models with web-collected image-text pairs[23, 26]. However, these texts may contain noise and fail to precisely describe the content of corresponding images (e.g., an image of a dog paired with the text "*The photo is taken on Sunday*"), and such positive pairs hinder the alignment process. Additionally, the current sampling is confined to local-level batches with a limited number of samples, thereby restricting the ability to sample optimal positives or negatives.

To address these problems, we propose the Global Positive-Negative Sampling (GPN-S) framework. As shown in Figure 2, GPN-S utilizes a pre-trained encoder, such as CLIP[25], to map image and text data from the entire dataset into a unified global embedding space. Within this space, GPN-S samples optimal positives and negatives by leveraging the neighboring relationships among samples. We will introduce how to calculate the neighboring relationships in section 3.2. Then, in section 3.3 and section 3.4, we

respectively give the detailed description for how GPN-S samples positives and negatives through neighboring relationships. The pseudocode is provided in the appendix.

## 3.2 Neighbor Calculating

GPN-S utilizes an off-the-shelf pre-trained model that consists of a visual encoder $g_V(\cdot)$ and a text encoder $g_T(\cdot)$. These encoders are employed to map image and text from the entire dataset into a unified global embedding space. Given a dataset with $D$ image-text pairs $\{(v_i, t_i)\}_{i=1}^D$, it computes the image embedding $\mathbf{v}_i^g = g_V(v_i)$ and text embedding $\mathbf{t}_i^g = g_T(t_i)$ for each pair. Then, we employ Faiss [14] to identify the $k$ nearest-neighbor texts for each image $v_i$, determined by the closest cosine distance to $v_i$'s embedding, denoted as $V2T_k(v_i)$:

$$V2T_k(v_i) = \{t_j \mid j \in \underset{j' \in [1, D]}{\text{top}_k} \{(\mathbf{v}_i^g)^T \mathbf{t}_{j'}^g\} \}, \quad (6)$$

where $\text{top}_k$ identifies the indices of the top $k$ largest values. Similarly, we compute $k$ nearest-neighbor images $V2V_k(v_i)$ for each image $v_i$, and $k$ nearest-neighbor images $T2V_k(t_i)$ for each text $t_i$. These cross-modal and uni-modal neighbors are crucial for guiding our sampling strategy.

## 3.3 Global Positive Sampling (GP-S)

Due to textual noise, the pair $(v_i, t_i)$ does not always serve as a genuinely positive pair that conducive to alignment. We address this issue from two perspectives. From cross-modal perspective, we construct a refined text aligned with $v_i$. From uni-modal perspective, we establish uni-modal alignment among sampled positive image

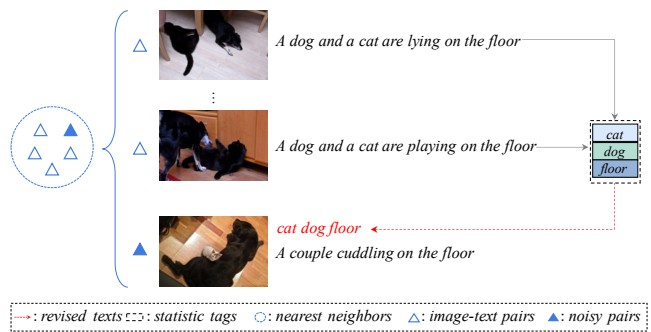

**Figure 3: Illustration of GP-S for cross-modal. We revise the original text based on nouns that frequently appear in neighboring texts.**

pairs, enabling the representation of $v_i$ to align with semantically similar text through other images. This section will elaborate on the utilization of global neighboring information for sampling positive cross-modal and uni-modal pairs.

*3.3.1 GP-S for Cross-Modal.* As mentioned earlier, positive sampling for cross-modal refers to constructing a refined text. We need to extract the most crucial visual information from the image, which pertains to the objects present in the image. We can infer the objects through the neighboring text, as shown in Figure 3. To be specific, we utilize SpaCy's part-of-speech tagging tool to extract all nouns from the dataset's text. Subsequently, we create a set of nouns, denoted as $S_n$, comprising those nouns that occur more than 10 times. For a given image $v_i$, if the frequency of a particular noun in its $k$ nearest-neighbor text surpasses a threshold $p$, we conclude that the image $v_i$ contains the object represented by that noun, denoted as $obj(v_i)$:

$$obj(v_i) = \{n | \frac{\sum_{t_j \in V2T_k(v_i)} \mathbb{1}_{(n \in t_j)}}{k} > p, n \in S_n\}, \quad (7)$$

where the binary indicator $\mathbb{1}_{(n \in t_j)}$ equals 1 if the noun $n$ appears in text $t_j$ and 0 otherwise. The values of $k$ and $p$ may vary depending on the specific characteristics of the dataset, and we can confirm these values by conducting a rapid manual check to ensure the extraction of objects from the image.

We transform $obj(v_i)$ by inserting spaces between the nouns, resulting in the revised text $t_i^r$. During training, it is crucial to determine which noise texts require replacement. We set a threshold $\alpha$. For image-text pairs $(v_i, t_i)$ with a similarity of $(\mathbf{v}_i^g)^T \mathbf{t}_i^g \leq \alpha$, we replace the text $t_i$ with the corresponding revised text $t_i^r$. Inspired by curriculum learning[4], we replace all text during epoch 0, i.e., $\alpha = \max_{i \in [1,D]} \{(\mathbf{v}_i^g)^T \mathbf{t}_i^g\}$. After half of the total training epochs, we do not replace any text, i.e., $\alpha \leq \min_{i \in [1,D]} \{(\mathbf{v}_i^g)^T \mathbf{t}_i^g\}$. We employ a linear decay for $\alpha$:

$$\alpha = \frac{e - E/2}{-E/2} \max_{i \in [1,D]} \{(\mathbf{v}_i^g)^T \mathbf{t}_i^g\} + \frac{e}{E/2} \min_{i \in [1,D]} \{(\mathbf{v}_i^g)^T \mathbf{t}_i^r\}, \quad (8)$$

where $E$ is the total number of epochs and $e$ is the number of the current epoch. This strategy ensures that the model is not disrupted by noise during the initial training, leading to a well-optimized

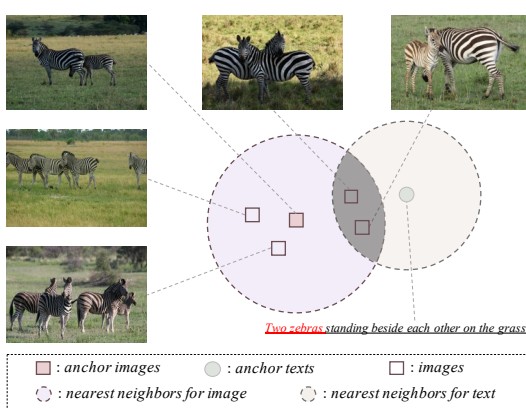

**Figure 4: Illustration of GP-S for uni-modal. We identify positive images through the intersection between $T2V$ and $V2V$, allowing more precise sampling.**

starting point. Once alignment performance is sufficiently high, the model can gradually comprehend more complex original text.

*3.3.2 GP-S for Uni-Modal.* For uni-modal alignment, the focus is to sample image pairs with both structural and semantic similarity, we define semantic-equivalent images $SE_i$ for $v_i$ as follows:

$$SE_i = V2V_{k_1}(v_i) \cap T2V_{k_2}(t_i), \quad (9)$$

where $k_1$ and $k_2$ are parameters that control the range of neighbors. Here, information from $V2V$ requires similarity in various aspects such as composition of the images, while information from $T2V$ places greater emphasis on the semantic similarity of the images. When there are no positive images for $v_i$ in the dataset, the intersection of the two sets would be empty, and we set $SE_i$ as a set containing randomly augmented versions of $v_i$.

In each training epoch, we randomly select a positive image $v_i^{\text{pos}}$ from $SE_i$ and compute the uni-modal contrastive loss defined as follows:

$$\mathcal{L}_{\text{uni}}(v_i, v_i^{\text{pos}}) = -log \frac{\exp((\mathbf{v}_i^{\text{cls}})^T \mathbf{v}_i^{\text{pos,cls}}/\tau)}{\sum_{j=1}^{N} \exp((\mathbf{v}_i^{\text{cls}})^T \mathbf{v}_j^{\text{cls}}/\tau)}, \quad (10)$$

where $\tau$ is a learnable temperature parameter, $N$ is the batch size.

For the input $(v_i, t_i, v_i^{pos})$, the final objective $\mathcal{L}$ is achieved by adding the uni-modal contrastive loss to the raw training loss of the model $\mathcal{L}_f$ outlined in its original paper:

$$\mathcal{L}(v_i, t_i, v_i^{\text{pos}}) = \mathcal{L}_f(v_i, t_i) + \mathcal{L}_{\text{uni}}(v_i, v_i^{\text{pos}}). \quad (11)$$

## 3.4 Global Negative Sampling (GN-S)

Previous works[2, 19, 25] have demonstrated the benefits of sampling hard negatives in VLP. However, in-batch mining is constrained by the batch size. In this section, we introduce global negative sampling, enabling us to acquire more challenging negative samples from the entire dataset.

An intuitive approach is to utilize $k$ nearest neighbors as hard negatives. However, to avoid potential positives in the negative neighbors, the value of $k$ needs to be set relatively high, which in

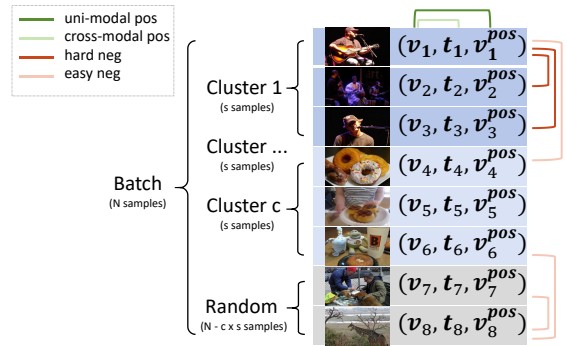

**Figure 5: Illustration of GN-S. We present an example of the composition of a batch, where $N = 8$, $c = 2$, and $s = 3$. Samples within the same cluster serve as hard negative pairs for each other.**

turn results in an increase in computational costs. To enhance efficiency, we opt for the $K$-means algorithm[12] for clustering, where any pair of samples within the same cluster serve as a hard negative pair. Specifically, we utilize Faiss[14] to partition the dataset into $K$ clusters, with each image-text pair $(v_i, t_i)$ represented by its image feature $\mathbf{v}_i^g$.

As recent VLP models have incorporated in-batch hard negative mining, introducing our GN-S only requires placing global-level hard negatives into the same batch. Specifically, when a batch of size $N$ is constructed, we randomly select $c$ clusters from all $K$ clusters and $s$ samples from each cluster. The remaining $N - c \times s$ samples are randomly chosen from the entire dataset, as shown in Figure 5. By employing this approach, the $s$ samples from the same cluster act as hard negatives for each other, resulting in a notable increased number of hard negatives compared to batches formed randomly. We observe that the model performs best when $c \times s \approx N/4$. This is done to maintain diversity within the batch, with still having $3N/4$ completely random samples, thus preventing the model from having difficulty distinguishing the 'easy' negatives.

### 3.5 Scalablity

Our framework is highly effective and efficient. Thanks to Faiss[14], computations such as 500-nearest-neighbor search and 1000-means clustering on a dataset containing 5 million image-text pairs can be completed within 2 hours, utilizing only 2 NVIDIA GeForce RTX-3090 GPUs. As a comparison, training the VLP model ALBEF[19] using 8 A100 GPUs takes 60 hours. **The additional computations introduced by our framework do not exceed 10% of the training time.** Regarding storage overhead, only the identifiers of the neighbors are stored, such as filenames. The storage space required is significantly smaller compared to that of the original image-text pairs.

When applying our framework to a large-scale dataset, we advise dividing the dataset randomly into groups of 5 million data pairs each and sampling neighbors only within each group. This approach has two advantages. First, our experiments have already demonstrated that the neighbor information computed on the 5 million dataset is already sufficient for our global positive and negative sampling, thereby obviating the need for any alteration in hyperparameters. Second, this processing method can be extended to datasets of any size, and the additional costs would also not exceed 10% of the training time.

## 4 EXPERIMENT

### 4.1 Implementation Details

Our GPN-S modifies the existing VLP model's sampling strategy to enhance their performance. We implement GPN-S on ALBEF [19] and TCL [32] model, referring to them as the backbone models. They share similar model architectures and pre-train datasets. Specifically, we utilized their base-size versions. The vision encoder in both models is a ViT-B/16 with 12 layers, and both the text encoder and the fusion encoder are implemented using a 6-layer transformer. The pre-training dataset includes COCO[21], Visual Genome[17], Conceptual Captions[27] and SBU Captions[24], comprising approximately 4 million images and 5.1 million image-text pairs.

By default, we use the backbone model as the off-the-shelf pre-trained encoder. For instance, when applying our methods to AL-BEF, we obtain the pre-trained checkpoint from their official GitHub repository for neighbor calculation. We empirically set the parameters as follows: $k = 10$, $p = 0.3$ for calculating $obj(v_i)$, $k_1 = 5$, $k_2 = 500$ for selecting positive image pairs, $c = 40$, $s = 3$ for constructing batches of size $N = 512$ in GN-S, with the number of clusters $K = 1000$. We maintain all settings and training details from the original backbone models, with one modification: we train TCL for 50 epochs, rather than 30, to achieve better convergence. All experiments are conducted on 8 NVIDIA A100 GPUs.

### 4.2 Downstream Tasks

To evaluate the effectiveness of our proposed method, we adapt the pre-trained model to various downstream vision-and-language (V+L) tasks, ensuring consistency in all settings with those of the backbone models[19, 32], as below:

*Image-Text Retrieval.* We evaluate both fine-tune and zero-shot retrieval performance on the MSCOCO[21] dataset and employ the widely used Karpathy split [15], which comprises 5000 images along with their corresponding 25010 texts. We consider two tasks: image-to-text retrieval (TR), which involves finding the corresponding text for a given image, and text-to-image retrieval (IR), which requires finding the corresponding image with a given text. The R@$n$ (Recall at $n$) measures the proportion of correct answers included within the top $n$ retrieved results. The RSUM metric is the sum of TR@1, TR@5, TR@10, IR@1, IR@5, and IR@10.

*Visual Question Answering (VQA).* VQA aims to answer natural language questions about images. We fine-tune our model on training and development set of VQAv2[1] and Visual Genome[17] VQA data, and evaluate on the test-dev and test-std set of VQAv2. We follow ALBEF and TCL, employing a decoder to generate answers from a pool of 3192 candidates.

*Natural Language for Visual Reasoning for Real (NLVR$^2$) [28].* NLVR$^2$ presents the task of determining whether a natural language

**Table 1: Results on fine-tune(FT) and zero-shot(ZS) retrieval tasks.**

| Method | MSCOCO-FT | | | | | | | MSCOCO-ZS | | | | | | |
|---|---|---|---|---|---|---|---|---|---|---|---|---|---|---|
| | TR@1 | TR@5 | TR@10 | IR@1 | IR@5 | IR@10 | RSUM | TR@1 | TR@5 | TR@10 | IR@1 | IR@5 | IR@10 | RSUM |
| CLIP[25] | - | - | - | - | - | - | - | 58.4 | 81.5 | 88.1 | 37.8 | 62.4 | 72.2 | 400.4 |
| ViLT[16] | 61.5 | 86.3 | 92.7 | 42.7 | 72.9 | 83.1 | 439.2 | 56.5 | 82.6 | 89.6 | 40.4 | 70.0 | 81.1 | 420.2 |
| UNITER[8] | 64.4 | 87.4 | 93.1 | 50.3 | 78.5 | 87.2 | 460.9 | 64.1 | 87.7 | 93.3 | 48.8 | 76.7 | 85.8 | 456.4 |
| CoCa[33] | - | - | - | - | - | - | - | 63.8 | 84.7 | 90.7 | 47.5 | 72.4 | 80.9 | 440.0 |
| VLMo[2] | 74.8 | 93.1 | 96.9 | 57.2 | 82.6 | 89.8 | 494.4 | - | - | - | - | - | - | - |
| METER[10] | 76.2 | 93.2 | 96.8 | 57.1 | 82.7 | 90.1 | 496.1 | - | - | - | - | - | - | - |
| ALBEF[19] | 73.1 | 91.4 | 96.0 | 56.8 | 81.5 | 89.2 | 488.0 | 68.7 | 89.5 | 94.7 | 50.1 | 76.4 | 84.5 | 463.9 |
| +ours | 76.0 | 92.9 | 96.7 | 58.5 | 82.7 | 89.6 | 496.4 | 71.8 | 91.1 | 95.2 | 53.8 | 79.3 | 87.1 | 478.3 |
| | (+2.9) | (+1.5) | (+0.7) | (+1.7) | (+1.2) | (+0.4) | (+8.4) | (+3.1) | (+1.6) | (+0.5) | (+3.7) | (+2.9) | (+2.6) | (+14.4) |
| TCL[32] | 75.6 | 92.8 | 96.7 | 59.0 | 83.2 | 89.9 | 497.2 | 71.4 | 90.8 | 95.4 | 53.5 | 79.0 | 87.1 | 477.2 |
| +ours | 77.3 | 93.7 | 96.8 | 59.9 | 83.6 | 90.2 | 501.5 | 72.5 | 91.8 | 95.7 | 55.3 | 80.5 | 88.0 | 483.8 |
| | (+1.7) | (+0.9) | (+0.1) | (+0.9) | (+0.4) | (+0.3) | (+4.3) | (+1.1) | (+1.0) | (+0.3) | (+1.8) | (+1.5) | (+0.9) | (+6.6) |

sentence is true about a pair of images. We follow ALBEF and TCL, extending the fusion encoder to enable reasoning over two images and adding a fully-connected layer to predict if the sentence is true or false.

*Visual Entailment (SNLI-VE) [31].* SNLI-VE is a novel inference task based on the Stanford Natural Language Inference corpus and Flickr30k dataset. Given a real world image premise and a natural language hypothesis, the goal is to determine if the hypothesis can be concluded given the information provided by image. We follow ALBEF and TCL, considering it as a three-classes classification problem because the relationship could be entailment, neutral or contradiction.

For VQA, NLVR$^2$ and SNLI-VE, we report the average accuracy on the test dataset.

### 4.3 Main Results

This subsection compares the performance of models with and without the GPN-S framework to demonstrate its effectiveness.

*4.3.1 Image-Text Retrieval.* We evaluate the performance of the model in both fine-tune and zero-shot settings, with the results shown in Table 1.

In both settings, the model incorporating GPN-S significantly outperforms the original backbone model across all metrics. Notably, the improvement is more pronounced in the zero-shot setting, with increases of +14.4(3.1%) for ALBEF and +6.6(1.4%) for TCL in terms of RSUM metrics. This enhancement indicates that GPN-S more effectively aligns images and text, and the more substantial increase in R@1 compared to R@5 and R@10 suggests that GPN-S enhances the model's ability to discern fine-grained differences.

*4.3.2 VQA, NLVR$^2$ and SNLI-VE.* We conduct fine-tuning on other downstream tasks requiring deep image-text interaction understanding. Table 2 shows the experimental results. Our method consistently outperforms the corresponding backbone models.

Specifically, on the VQA task, we achieve an average improvement of +0.52(+0.7%) and +0.61(+0.8%) for the ALBEF and TCL

**Table 2: Results on other vision-language tasks.**

| Method | VQA | | NLVR$^2$ | | SNLI-VE | |
|---|---|---|---|---|---|---|
| | test-dev | test-std | dev | test-P | val | test |
| ViLT | 71.26 | - | 75.70 | 76.13 | - | - |
| UNITER | 72.70 | 72.91 | 77.18 | 77.85 | 78.59 | 78.28 |
| UNIMO | 73.79 | 74.02 | - | - | 80.00 | 79.10 |
| VLMo | 76.64 | 76.89 | 82.77 | 83.34 | - | - |
| METER | 77.68 | 77.64 | 82.33 | 83.05 | 80.86 | 81.19 |
| ALBEF | 74.54 | 74.70 | 80.24 | 80.50 | 80.14 | 80.30 |
| +ours | 75.15 | 75.12 | 80.73 | 80.93 | 80.53 | 80.41 |
| | (+0.61) | (+0.42) | (+0.49) | (+0.43) | (+0.39) | (+0.11) |
| TCL | 74.90 | 74.92 | 80.54 | 81.33 | 80.51 | 80.29 |
| +ours | 75.46 | 75.58 | 81.14 | 81.44 | 80.63 | 80.51 |
| | (+0.56) | (+0.66) | (+0.60) | (+0.11) | (+0.12) | (+0.22) |

backbones, respectively. For the NLVR$^2$ task, our method outperforms the backbone by an average of 0.41(+0.5%). For the SNLI-VE task, our method yields an average improvement of 0.21(+0.3%) over the backbone. These gains are attributed solely to the improved pre-training alignment, as the fine-tuning process remains unchanged.

### 4.4 Ablation Study

*4.4.1 Sampling Strategies.* The effectiveness of each sampling strategy is validated through experiments, summarized in Table 3. Each strategy demonstrates improvement across all tasks, except SNLI-VE, compared to the backbone model. Combining two global positive sampling strategies yields better results than employing a single strategy, indicating that uni-modal and cross-modal positive samplings are complementary. Implementing all three strategies leads to the highest performance in most metrics, confirming their collective benefit.

*4.4.2 Off-The-Shelf Encoder.* We investigate the effectiveness of various off-the-shelf models. The results on the ALBEF backbone

**Table 3: Performance impact of individual sampling strategies on downstream tasks. For retrieval task, we report the RSUM metric.**

| Module | MSCOCO | | VQA | NLVR$^2$ | SNLI-VE |
|---|---|---|---|---|---|
| | ZS | FT | test-dev | test-P | test |
| ALBEF | 463.9 | 488.0 | 74.54 | 80.50 | 80.30 |
| +GP-S(Uni) | 474.0 | 492.7 | 74.74 | 80.74 | 80.25 |
| +GP-S(Cross) | 474.0 | 492.4 | 74.81 | 80.87 | 80.23 |
| +GP-S(Uni+Cross) | 475.1 | 492.8 | 75.07 | 80.91 | 80.36 |
| +GN-S | 475.9 | 493.0 | 75.05 | 80.90 | 80.28 |
| +GPN-S(FULL) | **478.3** | **496.4** | **75.15** | **80.93** | **80.41** |
| TCL | 477.2 | 497.2 | 74.90 | 81.33 | 80.29 |
| +GP-S(Uni) | 481.8 | 499.2 | 75.14 | 81.33 | 80.45 |
| +GP-S(Cross) | 481.7 | 499.3 | 75.21 | 81.35 | 80.52 |
| +GP-S(Uni+Cross) | 482.2 | 499.2 | 75.43 | 81.38 | **80.55** |
| +GN-S | 483.1 | 501.2 | 75.32 | 81.37 | 80.51 |
| +GPN-S(FULL) | **483.8** | **501.5** | **75.46** | **81.44** | 80.51 |

**Table 4: Performance impact of different off-the-shelf encoders. For retrieval task, we report the RSUM metric.**

| Encoder | MSCOCO | | VQAv2 | NLVR$^2$ | SNLI-VE |
|---|---|---|---|---|---|
| | ZS | FT | test-dev | test-P | test |
| - | 463.9 | 488.0 | 74.54 | 80.50 | 80.30 |
| CLIP | 477.1 | **496.4** | **75.27** | 81.11 | 80.23 |
| ALBEF | 478.3 | **496.4** | 75.15 | 80.93 | **80.41** |
| TCL | **479.1** | 496.3 | 75.12 | **81.20** | 80.22 |

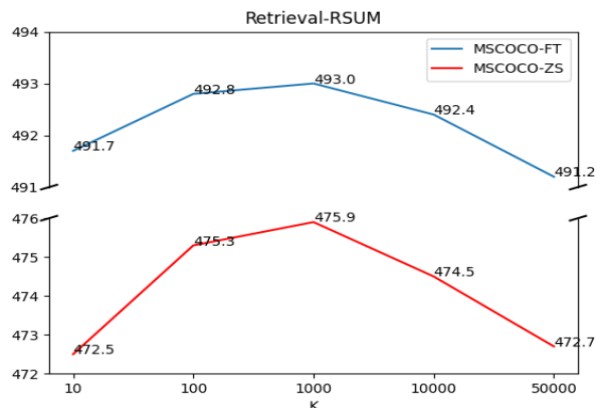

**Figure 6: Effectiveness of cluster number K in GN-S.**

are detailed in Table 4. By default, we use the backbone model that is not trained by our framework as the off-the-shelf model (so 'ALBEF' in the table denotes the default method). For comparison, we also utilize CLIP ViT-B/32 and TCL models as alternative off-the-shelf encoders, sourced from their official repositories. Our findings indicate that higher-performing off-the-shelf models yield better results in zero-shot retrieval tasks, while maintaining similar levels of performance in other downstream tasks. Given the significant

**Table 5: Effectiveness of modality used for clustering, cluster number $K = 1000$.**

| Modality | MSCOCO | | VQAv2 | NLVR$^2$ | SNLI-VE |
|---|---|---|---|---|---|
| | ZS | FT | test-dev | test-P | test |
| Vision | 475.9 | **493.0** | **75.05** | **80.90** | 80.28 |
| Text | **477.0** | 492.3 | 74.97 | 80.88 | **80.45** |

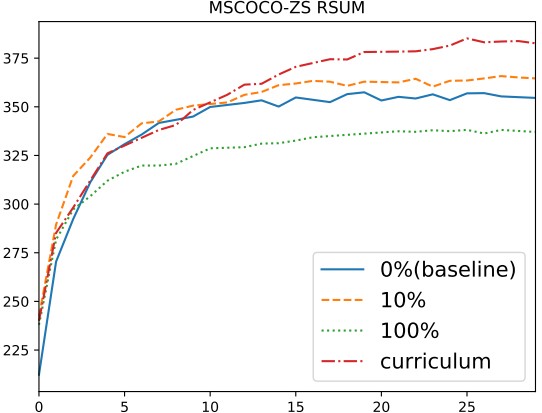

**Figure 7: The RSUM metric for zero-shot retrieval on the MSCOCO dataset at each epoch during the training process (x-axis represents the number of training epochs). The 0% line represents the original ALBEF model. The 10% line shows a fixed $\alpha$ scenarios where text is replaced if its similarity with the image falls within the lowest 10% of the dataset. The 100% line represents that all text is replaced. The curriculum line illustrates our curriculum-based scheduling for $\alpha$.**

improvements observed across all models, it is advisable to choose the best available off-the-shelf encoder for applying the GPN-S framework.

*4.4.3 Modality Used for Clustering.* We utilize $\mathbf{v}_i^g$ for image-text pair $(v_i, t_i)$ clustering by default, and we find that using either $\mathbf{v}_i^g$ or $\mathbf{t}_i^g$ can result in similar clustering outcomes, as the vision and text modalities have been aligned in the off-the-shelf model. Our experimental results on the 'ALBEF + GN-S' setting, presented in Table 5, further support our findings. We achieve comparable model performance through clustering with different modalities.

*4.4.4 Cluster Number K in GN-S.* In GN-S, we cluster all samples into $K$ clusters. As $K$ increases, the number of samples in each cluster decreases, resulting in higher similarity between samples and making it harder to distinguish them.

We select different values of $K$ and conduct an ablation study on ALBEF backbone, as presented in Figure 6. Optimal performance is achieved at $K = 1000$ on current dataset. Increasing $K$ further

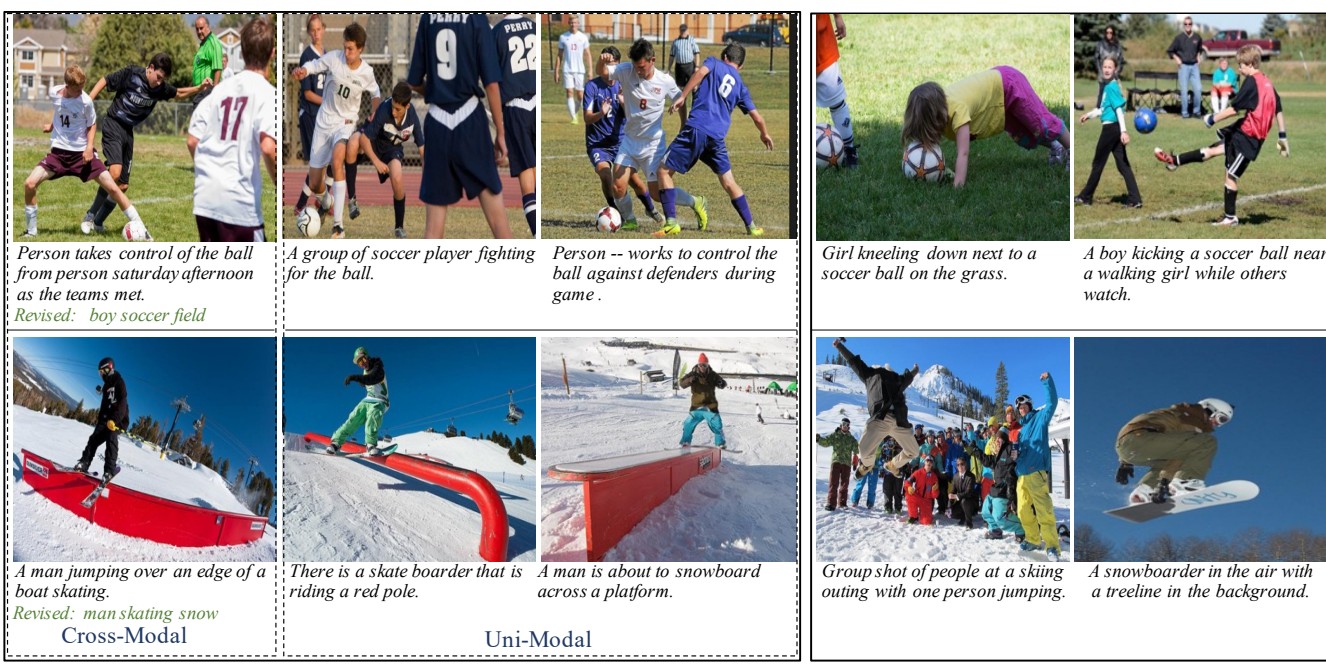

*Person takes control of the ball from person saturday afternoon as the teams met.*
*Revised: boy soccer field*

*A group of soccer player fighting for the ball.*

*Person -- works to control the ball against defenders during game .*

*Girl kneeling down next to a soccer ball on the grass.*

*A boy kicking a soccer ball near a walking girl while others watch.*

*A man jumping over an edge of a boat skating.*
*Revised: man skating snow*
Cross-Modal

*There is a skate boarder that is riding a red pole.*
Uni-Modal

*A man is about to snowboard across a platform.*

*Group shot of people at a skiing outing with one person jumping.*

*A snowboarder in the air with a treeline in the background.*

Positive Samples

Negative Samples

**Figure 8: Two examples are shown for visualization. The first column displays the original image-text pairs, while the following columns present the positive and negative samples obtained via our sampling strategy.**

results in performance drops, likely due to the issue of false negatives, where semantically similar samples are incorrectly treated as negatives.

## 4.5 GP-S for Cross-modal on Noisier Dataset

The GP-S method was proposed to address the impact of textual noise. However, in previous sections, we used relatively clean academic datasets for a fair comparison. In this section, we replace the pretraining dataset with a noisier random 3M subset of CC12M and use GP-S for cross-modal techniques without modifying the parameters $k$ and $p$. We report the RSUM metric for zero-shot retrieval on the MSCOCO dataset, as shown in Figure 7. We observe that on this dataset, we achieve a better improvement (RSUM improvement from 354.6 to 382.6, a 7.9% increase). In contrast to a fixed $\alpha$, curriculum learning for scheduling $\alpha$ proves to be the most effective approach. We believe that curriculum training, progressing from easy to hard, prevents the model from getting stuck in local optima.

As pretraining datasets in practical applications continue to grow, they will inevitably contain more noise. As the first work to introduce positive sampling in VLP, we hope this experiment demonstrates the importance of positive sampling in VLP.

## 4.6 Visualization of Positives and Negatives

For a clearer understanding of our sampling strategy, we provide visualizations of the positive and negative samples identified by GPN-S, as illustrated in Figure 8.

In the first example (first row), the text mentions *"saturday afternoon"* which is not visually represented in the image. We obtain

information through neighbors and revise the text accordingly. Moreover, We found neighbor images with similar semantic information and encouraged them to be close to each other in the embedding space. To distinguish hard negative samples within the same cluster, the model is required to fully comprehend the sentence *"takes control of the ball from people,"* rather than solely recognizing *"people"* and *"ball"*. In the second example, cross-modal positive sampling successfully captures the key information *"snow"* which is not explicitly mentioned in the original text. All positive images contain the infomation *"skating over a red platform"* while negative samples lack certain shared features with the anchor, yet are sufficiently challenging to enable the model to achieve fine-grained discrimination.

## 5 CONCLUSION

This paper introduces the Global Positive-Negative Sampling (GPN-S) framework, a novel framework that expands the sampling strategy of Vision-Language Pre-training (VLP) models to include both positive and negative pairs at a global level. By leveraging neighboring relations, GPN-S effectively identifies globally semantically similar neighbors as positive samples, enhancing the quality of representations in both cross-modal and uni-modal scenarios. Simultaneously, it encourages the model to differentiate between more challenging negative samples within the same cluster. Experimental results validate the effectiveness of this framework, demonstrating significant improvements for various VLP models on widely used benchmarks.

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
