# OpenReview forum: "Neighbor Does Matter: Global Positive-Negative Sampling for Vision-Language Pre-training"
_acmmm.org/ACMMM/2024/Conference — MM2024 Poster_

### Official Review · Reviewer_UAdT · 2024-05-21

**Rating:** 3
**Confidence:** 2

**Summary:**

This paper introduces a Global Positive-Negative Sampling (GPN-S) framework for vision-language pre-training. Unlike current methods that focus mainly on negative sampling at the local mini-batch level, GPN-S performs both positive and negative sampling on a global level based on neighborhood relationships. Experiments on several downstream tasks demonstrate that GPN-S significantly improves performance compared to existing models.

**Strengths:**

This paper introduces a Global Positive-Negative Sampling (GPN-S) framework for vision-language pre-training and evaluates it across several visual benchmarks to demonstrate performance. Overall, the paper is well-structured and easy to follow.

**Limitations:**

This paper addresses the limited scope of mini-batches or queues faced by previous methods and proposes performing sampling across the entire dataset. However, I think this approach could result in a significant computational burden. Therefore, it is necessary to compare the computational cost, as well as the training and inference time, between the current method and previous methods.

The authors compared various vision-language models, but the representative work BLIP[1,2] should also be included in the comparison.

Additionally, all the figures are not very clear and are difficult to understand, especially Figure 1.

[1] Li, J., Li, D., Xiong, C., & Hoi, S. (2022, June). Blip: Bootstrapping language-image pre-training for unified vision-language understanding and generation. In International conference on machine learning (pp. 12888-12900).
[2] Li, J., Li, D., Savarese, S., & Hoi, S. (2023, July). Blip-2: Bootstrapping language-image pre-training with frozen image encoders and large language models. In International conference on machine learning (pp. 19730-19742).

**Suitability:**

2

---

### Official Review · Reviewer_hmzt · 2024-05-24

**Rating:** 4
**Confidence:** 3

**Summary:**

In this paper, a Global Positive-Negative Sampling (GPN-S) is proposed to encompass both positive and negative pairs at a global level, thus significantly enhancing VLP model performance. In addition, this paper jointly considers both positive and negative samples for vision-language pre-training on the global-level perspective rather than a local-level mini-batch, which provides more informative and diverse samples.

**Strengths:**

This paper proposes a Global Positive-Negative Sampling (GPN-S) to encompass both positive and negative pairs at a global level, and conducts experiments to demonstrate the effectiveness of the GPN-S.
1. The paper is easy to understand and flows smoothly.
2. The relevant methods are explained clearly, and they clearly indicate the motivation and novelty.

**Limitations:**

1. As far as is known, the metrics for each task in Table 2 are greatly influenced by hyperparameters. The approximately 0.5 improvement mentioned in the text might be due to fluctuations during model training and testing, which is concerning.
2. Lines 512: "The additional computations introduced by our framework do not exceed 10% of the training time.", it is recommended to present additional comparative experimental results to demonstrate effectiveness.
3. What impact do N, c, and s in GN-S have on the experimental results? Are there any related experiments or discussions?
4. Please adjust the resolution of Figure 2 and Figure 6 for clearer readability by the readers.

**Suitability:**

3

---

### Official Review · Reviewer_paqj · 2024-05-24

**Rating:** 5
**Confidence:** 2

**Summary:**

This paper proposes a sampling strategy for VLP models that involves global positive and negative samples, differing from previous sampling methods. The authors validate their method on several vision-language downstream tasks, demonstrating its effectiveness.

**Strengths:**

1. The proposed sampling strategy is a novel strategy that operates on a global level, considering both positive and negative samples beyond the confines of mini-batches.
2. Combine with the proposed strategy, the results shows significant improvements over existing models on various downstream tasks.
3. The GPN-S framework is model-agnostic and can be effectively applied to large-scale datasets with a limited increase in computational costs.

**Limitations:**

1. How does the GPN-S handle the scalability issue when applied to extremely large datasets that may not fit into memory?
2. The authors used k-means to construct hard-negatives. However, as we know, k-means often encounters the curse of dimensionality when dealing with high-dimensional data, where some clusters may become very large while others are very small or even have zero samples, which clearly does not match the actual situation. How did the authors consider this issue?
3. Figure 6 is a bit blurry.
4. The authors should carefully check the format of the references.

**Suitability:**

3

---

### Official Review · Reviewer_YPEG · 2024-06-04

**Rating:** 4
**Confidence:** 2

**Summary:**

This paper introduces the Global Positive-Negative Sampling (GPN-S) framework for vision-language pre-training, addressing limitations in current sampling strategies. GPN-S performs both positive and negative sampling globally, based on neighborhood relationships, to bring semantically similar samples closer and push challenging negatives farther apart. Experiments on downstream tasks show significant performance improvements over existing models.

**Strengths:**

- Clear paper writing.

- Decent performance has been achieved.

- Reasonable motivation.

**Limitations:**

- The authors could supplement the paper with experiments related to classification, image captioning, and compositionality.

- It misses the comparison with other positive/negative sampling methods.

This work claims its biggest difference from previous related works is that it selects positive and negative samples from the entire dataset, rather than a mini-batch or mini-batch+memory-bank approach. However, I haven't seen any comparison provided.

**Suitability:**

3

---

### Meta-Review · Area_Chair_FGMi · 2024-06-29

**Recommendation:** Accept (Poster)
**Confidence:** 5

**Metareview:**

The submission proposed to leverage the more informative, positive samples that go beyond the local patch level to help along the VLP models. The submission has been been to consolidate with compelling evidences/justification that can favourably support the claims/contributions. Author responses clinched reviewers concerns to most. Some areas should be considered to improve the quality before publishing.

Critical experiments concerning broader experiments such as image captioning, classification and composition are missing. Those areas are test bed to evaluate any new VLP models. Failing to do so is less convincing to talk the audience around the idea.

The proposed method relies on global sampling spanning the entire dataset, which induces significant computational overheads. However, the authors seem to hedge this issue without providing substantiated proofs to buy this out, albeit the reviewers flagged this for elaborations.